# Market Efficiency and News Dynamics: Evidence from International Equity Markets

**Thomas C. Chiang** 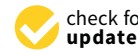

Department of Finance, LeBow College of Business, Drexel University, Philadelphia, PA 19104, USA; chiangtc@drexel.edu

**Abstract:** This paper examines the efficient market hypothesis by applying monthly data for 15 international equity markets. With the exceptions of Canada and the U.S., the null for the absence of autocorrelations of stock returns is rejected for 13 out of 15 markets. The evidence also rejects the independence of market volatility correlations. The null for testing the absence of correlations between stock returns and lagged news measured by lagged economic policy uncertainty (EPU) is rejected for all markets under investigation. The evidence indicates that a change of lagged EPUs positively predicts conditional variance.

**Keywords:** efficient market; economic policy uncertainty; random walk; news; Asian market; G7 market

---

## 1. Introduction

The aim of this paper is to present empirical evidence to evaluate the efficient market hypothesis (EMH) by using economic policy uncertainty (EPU) as a news variable. Specifying a regression model with longer lags of EPU allows us to test whether the EPU has a prolonged negative effect or being able to track a phenomenon for markets to rebound. A successful empirical finding from this study will help to inform investors of whether market damage is continually worsening or if they are to be rewarded by an uncertainty premium. Departing from conventional approaches that have focused on risk variables derived from financial market (Bali et al. 2009; Chen et al. 2018)[1], this study employs a broader definition of news variables by including consideration for EPU, which contains information of social event, political risk, or headline commentary news that can influence stock market. Thus, the results will be enriched by using EPU innovations to derive empirical regularities that can provide insight to investors about the stock return behavior.

In a highly integrated financial market system, a shock in one market will soon spillover to other markets. For instance, headline news on 10 October 2018 that the Nasdaq index suddenly dropped 4.08% in the U.S. market led to a corresponding 1.62% plunge in the UK FTSE and a drop of 1.09%, 1.54%, and 2.2.66%, in the German DAX 30, French CAC 40 and Russia MOER indices, respectively. The retreat in the U.S. caused declines in the Asian markets where China's Shanghai stock A-share plunged 5.22%, Hong Kong HSI fell 3.54%, and Japanese's Nikkei index was down 3.89%.

The above news observations provide us with some insight for analyzing the global market behavior. As news is released, regardless of whether its focus is financial or political, it will create

---

[1] This may stem from Frank Knight's statement (Knight 1921) regarding uncertainty that suggests economic agents have no historical data from which a probability distribution is developed. If there is any measure of uncertainty, which can be used as a proxy for the unexpected component of the state variable (Cornell 1983; Chiang 1985; Lauterbach 1989), then the omitted variable problem may arise.

uncertainty and hence fear among investors, regardless of whether the news comes from the domestic market or the foreign market. By employing the monthly data to test the news-based EPU indices (Baker et al. 2016; Davis 2016) on stock returns, this study finds that EPUs are positively correlated with stock returns beyond the current period, which allows us to reject the EMH and suggests the existence of an uncertainty premium. This is the case for nearly all markets, although the evidence is more consistent for the G7 markets. Further, with some minor exceptions, the evidence suggests that EPU innovations provide significant information that can be used to predict variance in a subsequent period. This finding leads us to reject the assumption that the error series is independently and identically distributed on monthly data. Following this introduction, Section 2 presents the essence of efficient market theory and sets up hypotheses to test the uncertainty premiums; Section 3 describes the data; Section 4 tests the monthly return autocorrelations. Section 5 presents a regression model to examine the effect of news on stock returns and reports the empirical results. Section 6 concludes the findings.

## 2. News and Market Efficient Market Theory

### 2.1. Efficient Market Hypothesis

In his Foundations of Finance (1976), Fama wrote: "An efficient capital market is a market that is efficient in processing information. The prices of securities observed at any time are based on 'correct' evaluation of all information available at that time. In an efficient market, prices 'fully reflect' available information." (Fama 1976, p. 134). This statement provides a central message regarding the importance of the timing of price information and market efficiency. Under this mechanism, stock prices provide an accurate signal to traders, which allows them to evaluate the value of firms. That is, firms can raise funds to finance their activities by selling securities at fair prices, and investors are able to acquire these assets at prices that fully reflect their underlying intrinsic values. In this sense, prices play a significant and effective function in allocating resources.

The essence of efficient market can be seen from its pricing process. This assumes that market participants at time $t-1$ use market information $\phi_{m,t-1}$ to assess a joint distribution of security prices for time $t$ as of $f_m\{p_{1t}, p_{2t}, \ldots, p_{nt} \text{I} \phi_{m,t-1}\}$, where $p_{it}$ is the price of security $i$ at time $t$ and $i = 1, 2, \ldots, n$. From this assessment of the distribution of prices, the market observes appropriate prices $\{p_{1t-1}, p_{2t-1}, \ldots, p_{nt-1}\}$ for individual securities, which in turn gives rise to the aggregate market price, $P_{t-1}$. Note that the information given by $\phi_{m,t-1}$ includes not only the prices per se, but also the process that describes the evolution of the state of the market over time. It follows that the market price, $P_{t-1}$, can be used to predict the expected price in $t$ given by:

$$E_{t-1}[P_t \text{ I } \phi_{m,t-1}] = P_{t-1}. \tag{1}$$

The above expression can be written as:

$$E_{t-1}[P_t - P_{t-1}\text{I } \phi_{m,t-1}] = E_{t-1}[\varepsilon_t\text{I } \phi_{m,t-1}] = 0. \tag{2}$$

This expression implies that errors in forecasting $P_t$ using $P_{t-1}$ on average approach zero. This also informs investors that if there are deviations by using $P_{t-1}$ in predicting the future stock price, $P_t$, the errors must be associated with random news that hits the market between time $t-1$ and $t$.[2]

Since price is sensitive to news, which arrives randomly in the market, the stock price can be said to wander along a random course. An analysis of an efficient market then tends to explore whether or not the market does, in fact, use available information in setting stock prices. One way to approach

---

2　This statement is based on Fama's perception (1976) of market efficiency. However, Ohlson (1995); Glezakos et al. (2012) and Jianu et al. (2014) find that financial statements provide a significant source information for predicting the stock price.

this issue is to explore whether the source of information for future price movements is associated with past prices. Let us consider that the one-period price evolves with a constant equilibrium rate, $\mu$, as:

$$P_t = P_{t-1}e^{(\mu+\varepsilon_t)} \tag{3}$$

Taking the natural logarithm using $p_t = ln(P_t)$, we obtain:

$$p_t = \mu + p_{t-1} + \varepsilon_t \tag{4}$$

where $\mu$ is the constant rate of an expected price change in the natural logarithm, $\varepsilon_t$ is independently and identically distributed (iid) with mean 0 and variance $\sigma^2$ and is expressed as $\varepsilon_t \sim$ iid (0, $\sigma^2$). The above expression in Equation (4) can be called a random walk with a drift, which can be alternatively expressed as:

$$R_{mt} = \mu + \varepsilon_t$$

where $R_{mt} = p_t - p_{t-1}$.

Taking expectation on prices, the notion of Equation (2) then can be expressed as:

$$E_{t-1}[R_{mt}\mathrm{I}\,\phi_{m,t-1}] - \mu = E_{t-1}[\varepsilon_t\mathrm{I}\,\phi_{m,t-1}] = 0. \tag{5}$$

Equation (5) suggests that an expected stock return deviates from its equilibrium is expected to be zero, which implies there are no systematic excess profits that can be explored by checking the evolution of stock returns over time. To test this proposition, a linear regression function can be used and expressed as:

$$E_{t-1}[R_{mt}\mathrm{I}\,\phi_{m,t-1}] = \mu + \rho_s R_{mt-s}. \tag{6}$$

It follows that market efficiency, in combination with the assumption of constant expected returns over time, implies an absence of autocorrelation of the returns with $s$ order of lags. Thus, the null hypothesis is: $\rho_s = 0$ for $s = 1, 2, \ldots, S$. A further test of market efficiency can be achieved by examining whether $\{E_{t-1}[R_{mt}\mathrm{I}\,\phi_{m,t-1}] - \mu\}$ is orthogonal to any lagged news.

In analyzing the market efficiency, researchers opt to apply the random walk process to describe the possible correlations of information dependency. Campbell et al. (1997) clearly distinguish the notion of independent assumption from that of a serial correlation and note that the assumption of incremental iid is too strong (Campbell et al. 1997), since even though $\mathrm{Cov}[\varepsilon_t, \varepsilon_{t-s}] = 0$ for all $s \neq 0$ cannot be rejected, $\mathrm{Cov}[\varepsilon_t^2, \varepsilon_{t-s}^2] = 0$ for some $s \neq 0$ is usually rejected. This has been displayed in stock return series where large returns tend to be followed by large returns, or the volatility of stock returns appears to be highly dependent, presenting a clustering phenomenon shown in most GARCH-type conditional variance (Ding et al. 1993; Chen and Chiang 2016; Chen et al. 2018).

*2.2. The Model with News Information*

Empirical analysis of the impact of news on stock returns follows two approaches. The first is to derive a news variable from rational expectations. Known as an efficient markets-rational expectations hypothesis (Mishkin 1982), it posits that investors are rational, and are able to use econometric models and available information to form an optimal forecast of a state variable. Thus, the unpredicted component is nothing but the result of news hitting market. This approach has been proposed by Mishkin (1982); Cornell (1983); Chiang (1985); Pearce and Roley (1985) and Lauterbach (1989). Clearly, this approach simply relies on a policy/state variable, such as an unexpected change in money supply or a change in interest rates as a proxy for measuring monetary policy uncertainty, which in turn is used to test the news impact on stock returns.

The second approach is based on survey data of market participants or headline news, which forms the future economic prospects that influence stock returns. For instance, McQueen and Roley (1993) find that good news in an industrial production index raises stock

prices. Boyd et al. (2005); Leduc and Liu (2016) and Caggiano et al. (2014) report that news of rising unemployment leads to contraction and lower expected earnings and hence results in lower stock prices. Birz and Lott (2011) choose newspaper articles as a measure of news. These authors indicate that news about GDP and unemployment affects stock returns.

Despite their success in linking macroeconomic news (Birz and Lott 2011) to the stock returns, their choice of news variables is restricted to macroeconomic indicators and fails to include broad coverage of news, such as the political risk, changes in immigration policy, and trade wars among others, which could significantly disrupt prevailing economic conditions and therefore affect investors' expectations and investment decisions. To alleviate the weakness arising from narrow news content, this study employs EPU indices which reflect broader news coverage as described in earlier. Further, most studies are focused on daily data; as a result, the impact of news has been treated as short lived. This approach ignores the longer-term effects on stock returns without investigating the delayed reactions to news. Finally, with the exception of Flannery and Protopapadakis (2002), very few studies pay attention to the issue of stock return heteroskedasticity. From an efficient market point of view, examining the dependency of volatility appears to be an integral part of analyzing investors' behavior.

Motivated by the above empirical issues, this study employs EPU indices to serve as news variables. The literature suggests that EPU affects both stock returns (Bansal et al. 2005; Ozoguz 2009; Antonakakis et al. 2013; Lopez de Carvalho 2017) and stock variance (Liu and Zhang 2015; Chiang 2019). To incorporate this notion into the test equations, we write:

$$R_{mt} = \alpha + \sum_{i=0}^{n} \beta_i \eta_{t-i} + \sum_{i=0}^{n} \gamma_i z_{t-i} + \rho_1 R_{m,t-1} + \varepsilon_t \tag{7}$$

$$\sigma_t^2 = \omega + b_1 \varepsilon_{(t-1)}^2 + b_3 \Delta \eta_{t-1} + b_4 \Delta z_{t-1} \tag{8}$$

where Equation (7) is the mean equation, $R_{mt}$ is the stock return, $\eta_t$ denotes the local $EPU_t$ and $z_t$ represents the global $EPU_t$. The AR(1) term is included in Equation (7) to capture either the momentum effect resulted from a price ceiling or the positive feedback of trading. Equation (8) is the variance equation, which assumes the GARCH(1,1)[3] process. However, the EPU innovations from respective local markets and the global market are included in the variance equation to capture the local news shock and contagious effect from global markets (Chiang et al. 2007; Forbes 2012; Bali and Cakici 2010). Finally, following Nelson (1991); Li et al. (2005); Chiang and Zhang (2018), the error series is assumed to follow the GED distribution, specified as $\varepsilon_t \Omega_{t-1} \sim \text{GED}(0, \sigma_t^2, \nu)$, which accommodates the thickness of the tails of a distribution.

*2.3. Uncertainty Premium Hypotheses*

Equation (7) provides a dynamic regression framework pertinent to test uncertainties and stock returns, this section outlines each hypothesis as follows:

(i)　　*Local uncertainty premium hypothesis*

If a rise in $\eta_{t-i}$ signifies a potential deterioration of economic activities that endangers future cash flows (Bloom 2009; Leduc and Liu 2016), it is expected that $E_{t-1}[R_{mt}, \eta_{t-i}] = 0$ would be rejected. Note a rejection of $E_{t-1}[R_{mt}, \eta_t] = 0$ is consistent with the EMH (Li 2017; Chen et al. 2017; Lopez de Carvalho 2017). However, if $E_{t-1}[R_{mt}, \eta_{t-i}] = 0$ for $i \geq 1$ is rejected and there is a positive relation,

---

3　　The popularity of this model is due to Bollerslev et al. (1992). Bollerslev (2010) provides different specifications of the conditional volatility models. In addition, some papers (Glosten et al. 1993; Chiang and Doong 2001) prefer to add an asymmetric term to the conditional variance equation to capture the bad news, which has a more profound impact on variance as compared to an equal amount of good news. Our specification indicates that this is redundant, since the inclusion of $\Delta \eta_{t-1}$ and $\Delta z_{t-1}$ already captures the effect arising from bad news.

then the market is inefficient and investors will be rewarded by an uncertainty premium from the local market.

(ii)    *Global uncertainty premium hypothesis*

It is observed that an increase in uncertainty over the global market is soon learned by local investors via mass media, digital devices, trade connections or financial institution linkages, which will induce investors to reassess their portfolio positions (Chiang et al. 2007; Forbes 2012; Klößner and Sekkel 2014; Chen et al. 2018). This spillover hypothesis can be tested by examining $E_{t-1}[R_{mt}, z_{t-i}] = 0$. A rejection of $E_{t-1}[R_{mt}, z_t] = 0$ is consistent with the efficient-market hypothesis. However, if $E_{t-1}[R_{mt}, z_{t-i}] = 0$ is rejected, that is, $\gamma_i = 0$ for $i \geq 1$ is rejected and $\gamma_i > 0$, evidence would go against the EMH, and investors will be rewarded by an uncertainty premium from a rise in lagged global EPU.

(iii)    *Uncertainty innovation hypothesis*

The literature suggests that uncertainty causes higher stock market volatility. Liu and Zhang (2015) show that the inclusion of EPU helps to improve forecasting ability of existing volatility models; and Tsai (2017) reports that EPU has a predictive ability not only to explain local stock volatility but also to describe cross market volatility. Testing these phenomena involves examining $\text{Cov}[\sigma_t^2, \Delta\eta_{t-1}] = 0$ and $\text{Cov}[\sigma_t^2, \Delta z_{t-1}] = 0$. In terms of Equation (8), the null hypothesis tests joint significance of $\Delta\eta_{t-1} = \Delta z_{t-1} = 0$ in a variance equation, which can be examined by Lagrange Multiplier (LM) test using the chi-squared distribution.

## 3. Description of Data and Variables

The empirical analyses in this study cover the data of the world stock index and 15 individual country/market indices, which include G7: Canada (CA), France (FR), Germany (GM), Italy (IT), Japan (JP), the United Kingdom (UK), the United States (US); Asian-Pacific markets: Australia (AU), China (CN), Hong Kong (HK), India (IN), South Korea (KO) and Singapore (SG); South American markets: Brazil (BR) and Chile (CL). Since most global EPU data start from January 1997, the estimations mainly use a sample for the period from January 1997 to June 2016. However, the stated times of stock indices for China and Brazil are later than other markets. The stock indices (including the total return index (RI) as defined in Datastream includes dividends, interest, rights offerings and other distributions realized over a given month) are downloaded from the database of DataStream, and the EPU news indices are obtained from www.PolicyUncertainty.com provided by Baker et al. (2016) and Davis (2016). The U.S. EPU index is constructed from three underlying components: (i) newspaper coverage of policy-related economic uncertainty based on major local newspapers; (ii) the number of tax code provisions set to expire in future years; (iii) disagreement among economic forecasters, which is used as a proxy for uncertainty. In constructing the EPU, Baker et al. (2016) search the digital archives of each paper to obtain a monthly count of articles that contain the following terms: "uncertainty" or "uncertain"; "economic" or "economy"; and one or more of the terms "deficit", "the Fed," or "uncertainties" or its variants. They find this uncertainty index is reliable, unbiased, and consistent since the uncertainty index is highly correlated with a market's implied volatility, VIX (Whaley 2009), and closely related to other measures of policy uncertainty. Following Bekaert and Harvey (1995), the stock prices are measured using the U.S. dollar.[4]

Table 1 reports summary statistics of monthly stock returns for the G7 market (Panel A) and Asian-Pacific and Latin American (APLA) markets (Panel B). The statistics indicate that the U.S. market performs well as compared with the other advanced markets; Japan, on the other hand, displays a negative return and high volatility as indicated by the standard deviation. The statistics in Panel B

---

[4]    Appendix A provides a description of a list of variables and data sources.

show that Chile, which has the highest return, performs very well, while China and South Korea, which have moderate returns, are relatively more volatile. In general, the returns in the group of APLA are much higher than those in G7 markets for the period of investigation.

**Table 1.** Summary statistics of monthly stock market returns: September 1990–June 2016.

| Panel A. G7 Market | | | | | | | | |
|---|---|---|---|---|---|---|---|---|
| | **CA** | **FR** | **GM** | **IT** | **JP** | **UK** | **US** | **Global** |
| Mean | 0.55 | 0.30 | 0.55 | 0.11 | −0.19 | 0.37 | 0.57 | 0.39 |
| Median | 0.62 | 0.90 | 0.95 | 0.04 | 0.24 | 0.66 | 1.00 | 0.78 |
| Maximum | 13.89 | 12.59 | 19.37 | 21.09 | 18.29 | 9.89 | 10.58 | 10.35 |
| Minimum | −25.53 | −19.23 | −29.33 | −16.80 | −27.22 | −13.95 | −18.56 | −21.13 |
| Std. Dev. | 5.56 | 5.30 | 6.02 | 6.00 | 6.02 | 3.95 | 4.08 | 4.23 |
| Skewness | −0.62 | −0.50 | −0.94 | 0.15 | −0.49 | −0.62 | −0.84 | −0.94 |
| Kurtosis | 4.77 | 3.58 | 6.22 | 3.80 | 4.40 | 3.93 | 5.17 | 5.56 |
| Jarque-Bera | 60.37 | 17.49 | 179.73 | 9.47 | 37.72 | 31.28 | 97.13 | 130.74 |
| Observations | 310 | 310 | 310 | 310 | 310 | 310 | 310 | 310 |
| Panel B. Asian-Pacific and Latin America markets | | | | | | | | |
| | **AU** | **CN** | **HK** | **IN** | **KO** | **SG** | **BZ** | **CL** |
| Mean | 0.79 | 0.75 | 0.90 | 1.11 | 0.67 | 0.50 | 1.11 | 1.27 |
| Median | 1.25 | 0.63 | 1.51 | 1.04 | 0.26 | 0.84 | 1.54 | 0.52 |
| Maximum | 9.73 | 92.34 | 25.30 | 53.79 | 42.89 | 21.33 | 20.54 | 17.44 |
| Minimum | −22.58 | −32.94 | −34.50 | −38.14 | −33.29 | −26.61 | −35.56 | −26.06 |
| Std. Dev. | 4.02 | 10.63 | 7.17 | 9.28 | 8.21 | 5.78 | 7.24 | 5.29 |
| Skewness | −0.89 | 2.33 | −0.30 | 0.13 | 0.41 | −0.47 | −0.81 | 0.14 |
| Kurtosis | 5.99 | 23.54 | 5.66 | 7.74 | 6.23 | 5.89 | 6.30 | 5.62 |
| Jarque-Bera | 156.67 | 4898.77 | 96.29 | 291.48 | 142.93 | 118.76 | 148.43 | 89.56 |
| Observations | 310 | 265 | 310 | 310 | 310 | 310 | 263 | 310 |

To visualize the stock returns, the monthly stock returns are plotted in Figures 1 and 2. These time series evidently present some degree of comovements and capture the major turning points, especially for the G7 markets, suggesting that these series could be driven by some common factors.

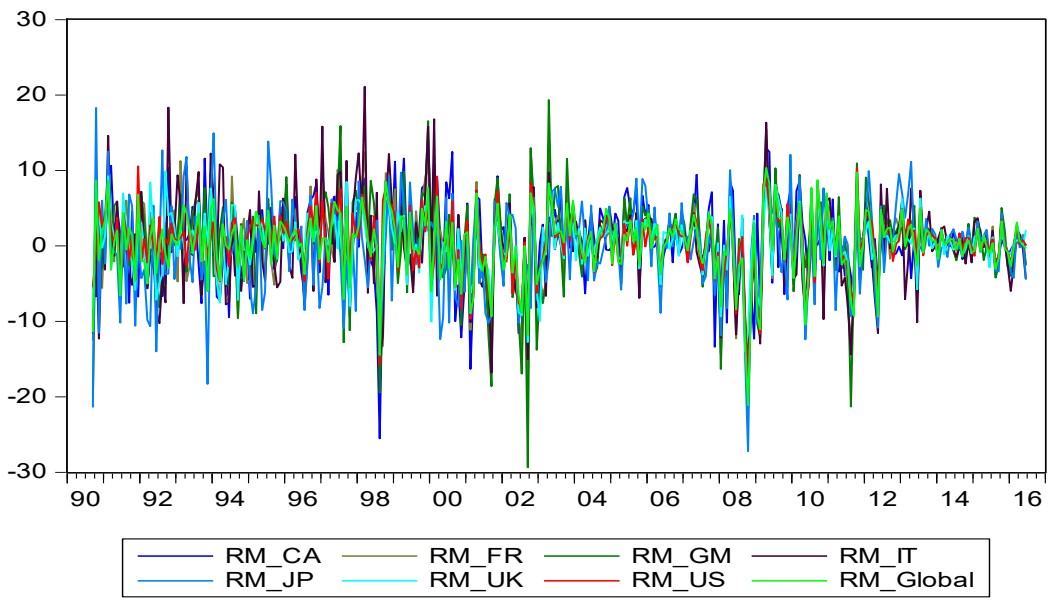

**Figure 1.** Time series plots of the percentage of stock returns (vertical axis) vs. time for G7 and global markets.

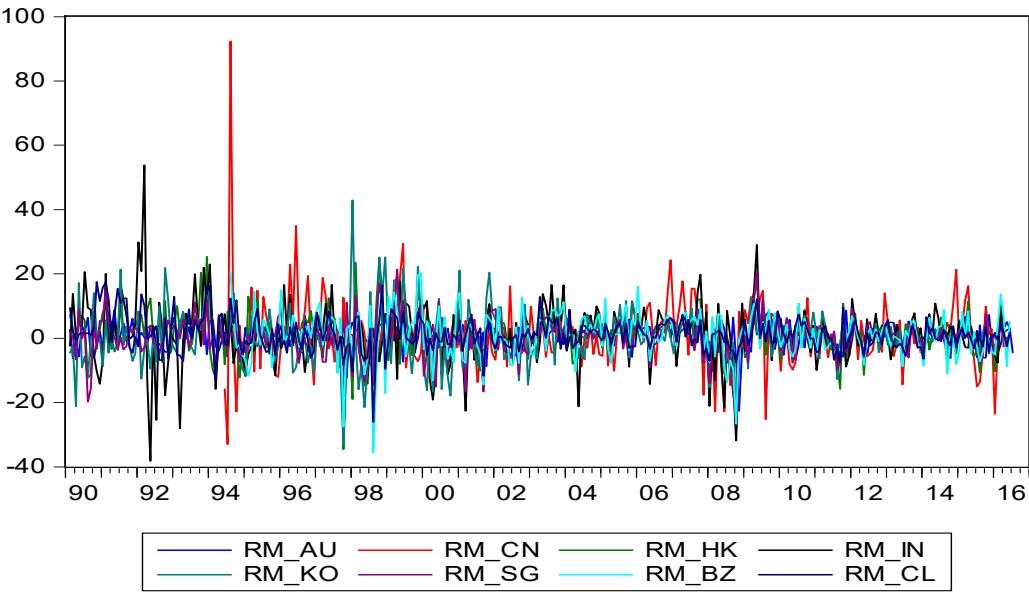

**Figure 2.** Time series plots of the percentage of stock returns (vertical axis) vs. time for Asia-Pacific and Latin American markets.

Let us turn to the EPU series, which are plotted in Figure 3 for the G7 market and Figure 4 for the APLA markets. The time paths of G7 markets exhibit some degree of comovements over time, and their correlations with global EPU (GEPU) are in the range from 0.48 (for Italy) to 0.88 (for the U.S.). It is evident that the EPU index for the UK spiked during the time of Brexit. Similarly, correlation coefficients of EPUs shown in the APLA group in Figure 4 range from 0.67 (for India) to 0.98 (for Singapore); the high correlation may reflect some sort of global contagions as shocks occur in the global markets (Chiang et al. 2007; Forbes 2012). The time paths show that EPUs for China, Hong Kong and South Korea occasionally act more volatile.

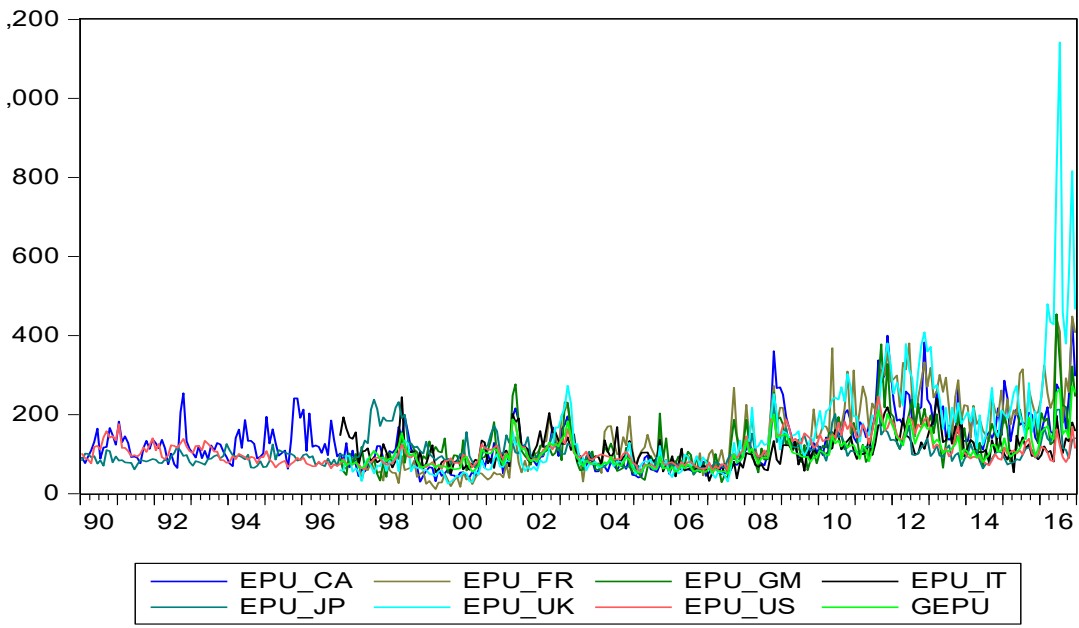

**Figure 3.** Time series plots of EPU (vertical axis) vs. time for G7 and global markets.

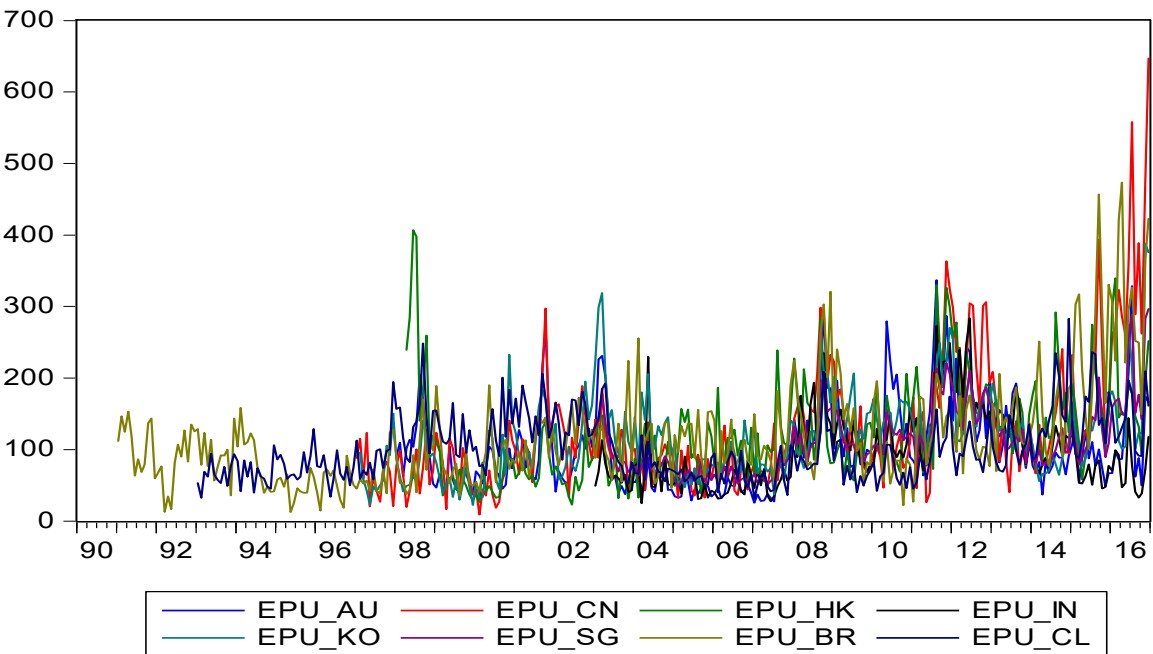

**Figure 4.** Time series plots of EPU (vertical axis) vs. time for Asia-Pacific and Latin American markets.

## 4. Test of Return Autocorrelations

A simple approach to testing the EMH is to examine the autocorrelations at the lagged *s* periods by *t*-statistics or examine the joint significance for $\rho_s = 0$ for $s = 1, 2, \ldots, S$ by $\chi^2$ distribution. Tables 2 and 3 report the autocorrelations of monthly stock market returns up to 12 orders for the G7 and APLA markets, respectively. Interestingly, only Canada and the U.S. lack autocorrelations, regardless of whether testing is on the individual lags or the lagged coefficients as a group. The results for Canada and the U.S. seem consistent with the EMH if we just look at the autocorrelation pattern. For the other markets, the evidence shows that at least some autocorrelations are significant, which leads to a rejection of the EMH. However, conclusions for the other G7 markets, with the exception of Italy, should be made with caution, since they do not display a consistent pattern, but rather exhibit significant autocorrelations in higher orders. This may result from a spurious correlation due to an omitted variable. If we look at the autocorrelations of the APLA markets in Table 3, five markets are statistically significant at the AR(1) term, which could result from price ceilings imposed by government or some sort of market frictions.

In testing whether the volatility is independent, that is, $\mathrm{Cov}[\varepsilon_t^2, \varepsilon_{t-s}^2] = 0$ for $s \neq 0$ as noted by Campbell et al. (1997), Table 4 reports the autocorrelations for absolute values of $R_{mt}$. Apparently, the null is uniformly rejected, as evidenced by the level of significance of the Ljung-Box Q-statistics up to 12 lags. Thus, the testing results show an absence of an independent assumption for volatility is invalid and suggest some type of GARCH model may be considered to describe the return residuals.

**Table 2.** Autocorrelations of monthly stock market returns up to 12 orders: Group 7 markets.

| Market | $\rho_1$ | $\rho_2$ | $\rho_3$ | $\rho_4$ | $\rho_5$ | $\rho_6$ | $\rho_7$ | $\rho_8$ | $\rho_9$ | $\rho_{10}$ | $\rho_{11}$ | $\rho_{12}$ | $Q_{12}$ |
|---|---|---|---|---|---|---|---|---|---|---|---|---|---|
| CA | 0.086 | 0.097 | 0.092 | −0.002 | −0.081 | −0.062 | −0.017 | 0.100 | 0.046 | 0.018 | 0.047 | −0.081 | 18.58 |
|  | 1.47 | 1.64 | 1.55 | −0.03 | −1.37 | −1.06 | −0.28 | 1.70 | 0.78 | 0.30 | 0.81 | −1.38 | [0.10] |
| FR | 0.125 | −0.081 | 0.095 | 0.029 | −0.006 | 0.025 | −0.076 | 0.146 | −0.072 | 0.110 | −0.050 | 0.101 | 13.18 |
|  | 2.13 * | −1.37 | 1.62 | 0.49 | −0.10 | 0.43 | −1.32 | 2.53 * | −1.23 | 1.90 | −0.86 | 1.77 | [0.36] |
| GM | 0.068 | 0.006 | 0.045 | 0.035 | 0.017 | 0.037 | −0.079 | 0.098 | −0.049 | 0.005 | −0.004 | 0.116 | 10.00 |
|  | 1.17 | 0.11 | 0.76 | 0.60 | 0.29 | 0.64 | −1.36 | 1.68 | −0.83 | 0.09 | −0.06 | 2.03 * | [0.62] |
| IT | 0.036 | 0.004 | 0.120 | 0.099 | −0.111 | −0.004 | −0.119 | 0.148 | 0.099 | −0.019 | 0.034 | 0.040 | 23.46 |
|  | 0.61 | 0.06 | 2.03 * | 1.68 | −1.90 | −0.07 | −2.05 * | 2.55 * | 1.70 | −0.33 | 0.58 | 0.70 | [0.02] * |
| JP | 0.086 | −0.021 | 0.103 | 0.032 | 0.004 | −0.128 | −0.027 | 0.053 | 0.028 | 0.038 | −0.012 | −0.037 | 7.36 |
|  | 1.46 | −0.35 | 1.76 | 0.54 | 0.07 | −2.17 * | −0.45 | 0.91 | 0.48 | 0.66 | −0.22 | −0.67 | [0.83] |
| UK | 0.044 | −0.034 | −0.022 | 0.136 | −0.015 | −0.011 | 0.005 | 0.068 | 0.015 | −0.003 | −0.024 | 0.036 | 12.63 |
|  | 0.75 | −0.58 | −0.38 | 2.31 * | −0.26 | −0.18 | 0.08 | 1.16 | 0.25 | −0.05 | −0.42 | 0.62 | [0.40] |
| US | 0.073 | −0.017 | 0.105 | 0.047 | 0.052 | −0.079 | 0.051 | 0.035 | −0.010 | 0.010 | 0.038 | 0.070 | 11.47 |
|  | 1.24 | −0.29 | 1.79 | 0.80 | 0.88 | −1.34 | 0.87 | 0.60 | −0.17 | 0.18 | 0.66 | 1.20 | [0.49] |

Notes: This table examines stock market efficiency by testing the dependency of stock returns; $\rho_s$ is the coefficient of autocorrelation with order $s$ ($s = 1, 2, \ldots, 12$). $Q_{12}$ is the Ljund-Box statistics for testing joint significance of 12 order lags. The numbers in the brackets are the *p*-values. The values in the first row are the estimated coefficients and in the second row are the *t*-statistics. * indicates statistically significant at the 5% level.

**Table 3.** Autocorrelations of monthly stock market returns up to 12 orders: Asian-Pacific and Latin American markets.

| Market | $\rho_1$ | $\rho_2$ | $\rho_3$ | $\rho_4$ | $\rho_5$ | $\rho_6$ | $\rho_7$ | $\rho_8$ | $\rho_9$ | $\rho_{10}$ | $\rho_{11}$ | $\rho_{12}$ | $Q_{12}$ |
|---|---|---|---|---|---|---|---|---|---|---|---|---|---|
| AU | 0.002 | 0.079 | 0.130 | 0.041 | −0.062 | −0.045 | 0.126 | 0.019 | 0.061 | −0.035 | −0.056 | 0.061 | 8.68 |
|  | 0.04 | 1.35 | 2.23 | 0.70 | −1.05 | −0.77 | 2.17 * | 0.33 | 1.04 | −0.61 | −0.96 | 1.04 | [0.73] |
| CN | 0.131 | 0.136 | −0.059 | 0.123 | 0.033 | −0.138 | 0.073 | 0.002 | 0.002 | −0.041 | 0.068 | −0.024 | 9.44 |
|  | 2.05 * | 2.12 * | −0.92 | 1.90 | 0.51 | −2.11 * | 1.11 | 0.03 | 0.04 | −0.79 | 1.33 | −0.47 | [0.67] |
| HK | 0.084 | 0.037 | 0.004 | −0.032 | 0.023 | −0.019 | 0.132 | 0.026 | 0.050 | 0.049 | −0.110 | −0.076 | 21.25 |
|  | 1.44 | 0.64 | 0.06 | −0.56 | 0.41 | −0.32 | 2.32 * | 0.45 | 0.87 | 0.85 | −1.90 | −1.32 | [0.05] * |
| IN | 0.121 | 0.050 | −0.010 | −0.068 | 0.097 | 0.061 | −0.046 | −0.034 | −0.038 | 0.003 | 0.043 | −0.083 | 16.96 |
|  | 2.08 * | 0.86 | −0.17 | −1.16 | 1.66 | 1.03 | −0.79 | −0.58 | −0.65 | 0.05 | 0.75 | −1.45 | [0.15] |
| KO | 0.152 | −0.069 | 0.030 | −0.092 | 0.030 | 0.010 | 0.040 | −0.016 | 0.061 | −0.063 | 0.060 | −0.084 | 9.75 |
|  | 2.61 * | −1.18 | 0.52 | −1.58 | 0.52 | 0.18 | 0.69 | −0.28 | 1.07 | −1.12 | 1.05 | −1.49 | [0.64] |
| SG | 0.120 | 0.142 | −0.043 | 0.058 | −0.031 | −0.037 | 0.036 | 0.007 | −0.005 | −0.042 | −0.091 | 0.042 | 17.38 |
|  | 2.06 * | 2.43 * | −0.73 | 0.99 | −0.53 | −0.64 | 0.63 | 0.12 | −0.08 | −0.74 | −1.60 | 0.74 | [0.14] |
| BR | 0.089 | 0.046 | 0.033 | 0.108 | −0.099 | −0.069 | 0.057 | 0.003 | −0.003 | 0.124 | 0.026 | −0.013 | 15.17 |
|  | 1.38 | 0.71 | 0.50 | 1.66 | −1.52 | −1.06 | 0.87 | 0.04 | −0.04 | 1.93 | 0.41 | −0.20 | [0.23] |
| CL | 0.201 | −0.012 | −0.005 | 0.170 | −0.036 | 0.009 | 0.148 | 0.057 | 0.041 | −0.025 | −0.017 | 0.014 | 43.84 |
|  | 3.48 * | −0.20 | −0.09 | 2.93 * | −0.61 | 0.15 | 2.53 * | 0.96 | 0.70 | −0.43 | −0.29 | 0.25 | [0.00] * |

Notes: This table tests the dependency of stock returns; $\rho_s$ is the coefficient of autocorrelation up to the 12th order. $Q_{12}$ is the Ljund-Box statistics for testing joint significance of 12 order lags. The numbers in the brackets are the *p*-values. The values in the first row are the estimated coefficients and in the second row are the *t*-statistics. * indicates statistically significant at the 5% level or better.

**Table 4.** Autocorrelations of monthly absolute values of stock market returns up to 12 orders.

| Market | $\rho_1$ | $\rho_2$ | $\rho_3$ | $\rho_4$ | $\rho_5$ | $\rho_6$ | $\rho_7$ | $\rho_8$ | $\rho_9$ | $\rho_{10}$ | $\rho_{11}$ | $\rho_{12}$ | Q(12) | P(s) |
|---|---|---|---|---|---|---|---|---|---|---|---|---|---|---|
| Panel A | | | | | | | | | | | | | | |
| CA | 0.231 | 0.291 | 0.255 | 0.149 | 0.269 | 0.266 | 0.215 | 0.276 | 0.170 | 0.168 | 0.218 | 0.142 | 195.88 | 0.00 |
| FR | 0.203 | 0.242 | 0.189 | 0.092 | 0.164 | 0.170 | 0.142 | 0.085 | 0.147 | 0.124 | 0.064 | 0.133 | 90.84 | 0.00 |
| GM | 0.124 | 0.222 | 0.195 | 0.084 | 0.167 | 0.191 | 0.123 | 0.098 | 0.169 | 0.148 | 0.046 | 0.118 | 84.33 | 0.00 |
| IT | 0.134 | 0.140 | 0.241 | 0.090 | 0.076 | 0.174 | 0.088 | 0.113 | 0.139 | 0.068 | 0.030 | 0.079 | 66.36 | 0.00 |
| JP | 0.147 | 0.101 | 0.101 | 0.101 | 0.098 | 0.019 | 0.043 | 0.047 | 0.110 | 0.032 | 0.040 | 0.050 | 26.49 | 0.01 |
| UK | 0.177 | 0.177 | 0.222 | 0.117 | 0.155 | 0.153 | 0.101 | 0.090 | 0.091 | 0.132 | 0.038 | 0.005 | 69.11 | 0.00 |
| US | 0.235 | 0.213 | 0.215 | 0.241 | 0.231 | 0.216 | 0.175 | 0.110 | 0.089 | 0.182 | 0.050 | 0.125 | 128.92 | 0.00 |
| Panel B | | | | | | | | | | | | | | |
| Au | 0.081 | 0.152 | 0.064 | 0.055 | 0.108 | 0.003 | 0.033 | −0.036 | 0.119 | 0.045 | 0.052 | 0.012 | 22.16 | 0.04 |
| CN | 0.142 | 0.159 | 0.048 | 0.087 | 0.105 | 0.044 | 0.13 | 0.041 | 0.124 | 0.1 | 0.117 | 0.034 | 34.69 | 0.00 |
| HK | 0.049 | 0.118 | 0.162 | 0.126 | 0.099 | 0.15 | 0.16 | 0.19 | 0.086 | 0.03 | 0.125 | 0.11 | 59.96 | 0.00 |
| IN | 0.109 | 0.328 | 0.101 | 0.119 | 0.202 | 0.101 | 0.139 | 0.084 | 0.018 | 0.073 | 0.031 | 0.155 | 79.69 | 0.00 |
| KO | 0.053 | 0.19 | 0.265 | 0.189 | 0.219 | 0.209 | 0.169 | 0.163 | 0.29 | 0.092 | 0.175 | 0.127 | 137.45 | 0.00 |
| SG | 0.227 | 0.122 | 0.081 | 0.091 | 0.102 | 0.207 | 0.222 | 0.183 | 0.155 | 0.100 | 0.024 | 0.049 | 80.73 | 0.00 |
| BZ | 0.148 | 0.094 | 0.125 | 0.112 | 0.065 | 0.036 | 0.152 | 0.124 | 0.095 | 0.125 | 0.114 | 0.088 | 40.29 | 0.00 |
| CL | 0.247 | 0.167 | 0.25 | 0.183 | 0.123 | 0.164 | 0.331 | 0.133 | 0.093 | 0.082 | 0.004 | 0.085 | 119.24 | 0.00 |

Notes: The standard error for the estimated coefficient is $1/\sqrt{T} = 1/\sqrt{310} = 0.05679$, e.g., the *t*-statistic of $\rho_1$ for CA is $0.231/0.05679 = 4.068$. Testing for absence of autocorrelations of the absolute $R_{mt}$ with 12 lags by Q(12), the null is uniformly rejected at the 1% level for all markets shown in *p*-values.

## 5. Empirical Results

### 5.1. Evidence from the Regression Method

The regression results of Equation (7), which use the Newey–West estimator (Newey and West 1987), are reported in Tables 5 and 6 that show negative and statistically significant values for all the coefficients of news variables, except those for Australia and Singapore in the global news at the current period. In the case of Australia, however, the negative impact has been extended to the lagged one period. The testing results also show that none of the AR(1) terms is significant. The evidence seems inconsistent with our earlier finding in testing the autocorrelations. In fact, this may be attributable to the fact that the AR(1) effect has been picked up by the lagged news variables in a multivariate regression procedure. In addition, the use of the Newey–West method also helps to reduce the significance of AR(1). However, the significance of the lagged news variables can be viewed as evidence against the EMH.

**Table 5.** Regression results of stock returns on domestic EPU ($\eta_t$) and global EPU ($z_t$) for G7 markets for the period of January 1997–June 2016.

| Markets | $\alpha$ | $\eta_t$ | $\eta_{t-1}$ | $\eta_{t-2}$ | $z_t$ | $z_{t-1}$ | $z_{t-2}$ | $\rho_1$ | $\overline{R}^2$ |
|---|---|---|---|---|---|---|---|---|---|
| CA | 1.751 | −0.042 | 0.009 | 0.024 | −0.082 | 0.052 | 0.020 | 0.05 | 0.11 |
|    | 1.87 | −3.57 | 0.78 | 2.38 | −3.46 | 2.00 | 0.90 | 0.71 | |
| FR | 0.745 | −0.029 | 0.019 | 0.006 | −0.111 | 0.073 | 0.016 | 0.01 | 0.14 |
|    | 1.11 | −4.27 | 2.34 | 0.90 | −5.60 | 2.91 | 0.80 | 0.14 | |
| GM | 0.676 | −0.072 | 0.046 | 0.024 | −0.130 | 0.059 | 0.033 | −0.07 | 0.16 |
|    | 0.63 | −5.14 | 2.40 | 1.96 | −5.09 | 1.73 | 1.24 | −1.08 | |
| IT | −0.198 | −0.070 | 0.031 | 0.041 | −0.093 | 0.063 | 0.003 | −0.07 | 0.16 |
|    | −0.16 | −5.34 | 1.91 | 3.09 | −5.05 | 2.56 | 0.14 | −1.03 | |
| JP | 2.006 | −0.089 | 0.029 | 0.043 | −0.062 | 0.032 | 0.043 | −0.02 | 0.19 |
|    | 1.92 | −4.47 | 1.34 | 2.57 | −3.85 | 1.24 | 1.81 | −0.30 | |
| UK | −0.334 | −0.017 | 0.022 | −0.001 | −0.081 | 0.037 | 0.015 | −0.09 | 0.15 |
|    | −0.75 | −2.42 | 2.48 | −0.21 | −4.98 | 1.77 | 0.95 | −1.27 | |
| US | 0.975 | −0.053 | 0.020 | 0.030 | −0.053 | 0.043 | 0.034 | −0.02 | 0.12 |
|    | 1.11 | −5.22 | 1.16 | 2.38 | −1.85 | 1.49 | 1.37 | −0.18 | |

Notes: The dependent variable is stock return. The values in the first row are the estimated coefficients and in the second row are the *t*-statistics.

**Table 6.** Regression results of stock returns on domestic EPU ($\eta_t$) and global EPU ($z_t$) for G7 markets for the period of January 1997–June 2016.

| Markets | $\alpha$ | $\eta_t$ | $\eta_{t-1}$ | $\eta_{t-2}$ | $z_t$ | $z_{t-1}$ | $z_{t-2}$ | $\rho_1$ | $\overline{R}^2$ |
|---|---|---|---|---|---|---|---|---|---|
| AU | 2.079 | −0.010 | −0.039 | 0.035 | −0.007 | −0.042 | 0.054 | −0.089 | 0.15 |
|    | 3.71 | −1.04 | −2.46 | 5.64 | −0.33 | −1.93 | 3.14 | −0.96 | |
| CN | 0.848 | −0.026 | 0.005 | 0.018 | −0.076 | 0.030 | −0.009 | 0.058 | 0.03 |
|    | 0.90 | −2.44 | 0.33 | 1.61 | −2.44 | 0.92 | −0.22 | 0.86 | |
| HK | 1.539 | −0.038 | 0.009 | 0.022 | −0.087 | 0.068 | 0.009 | 0.052 | 0.13 |
|    | 1.62 | −4.45 | 0.68 | 2.15 | −3.40 | 1.99 | 0.38 | 0.72 | |
| IN | 3.956 | −0.080 | 0.024 | 0.030 | −0.017 | −0.058 | 0.073 | 0.003 | 0.17 |
|    | 3.16 | −4.16 | 1.10 | 1.43 | −0.56 | −1.45 | 1.82 | 0.03 | |
| KO | −0.605 | −0.055 | 0.039 | 0.029 | −0.079 | 0.057 | 0.019 | 0.090 | 0.06 |
|    | −0.33 | −3.32 | 1.46 | 1.65 | −2.79 | 1.35 | 0.62 | 1.59 | |
| SG | 2.346 | −0.057 | 0.003 | 0.041 | −0.087 | 0.051 | 0.057 | 0.159 | 0.15 |
|    | 2.19 | −2.80 | 0.10 | 1.72 | −1.26 | 0.71 | 0.59 | 1.53 | |
| BR | 0.862 | −0.027 | 0.012 | 0.016 | −0.095 | 0.049 | 0.022 | −0.023 | 0.06 |
|    | 0.80 | −2.84 | 1.04 | 1.73 | −3.89 | 1.51 | 0.88 | −0.33 | |
| CL | 1.109 | −0.035 | 0.007 | 0.025 | −0.034 | 0.025 | 0.002 | −0.006 | 0.08 |
|    | 1.48 | −2.95 | 0.52 | 2.37 | −2.11 | 1.23 | 0.12 | −0.07 | |

Notes: The values in the first row are the estimated coefficients and in the second row are the *t*-statistics.

## 5.2. GARCH(1,1)-X Method

The estimated results reported in Tables 5 and 6 omit the terms of $\Delta\eta\_(t-1)$ and $\Delta z\_(t-1)$, implyingthe ignorance of EPU innovations on conditional variance. The system Equations (7) and (8) address this issue, and the estimated results using GED-GARCH(1,1)-M are reported in Tables 7 and 8. Several important findings are now summarized. First, in examining the coefficient of AR(1), autocorrelations are not significant in the G7 markets. However, autocorrelations for four markets in the Pacific-Asian group, including Australia, China, South Korea and Singapore, are significant. Viewed from this perspective, the G7 markets appear to behave more consistently with market efficiency than other markets do.

Second, the coefficients for the news variables at the current period, $\eta_t$ and $z_t$, are all negative and statistically significant; the exception is the U.S. market where the coefficient of global news is insignificant. This evidence, which shows current news variables are significant, does not go against market efficiency. However, in checking the lagged news, either one of $\eta_{t-1}, \eta_{t-2}, z_{t-1}, z_{t-2}$ or combinations of these lagged variables, we find the null should be rejected, indicating a lack of market efficiency. However, the patterns among the G7 markets (except for CA in the $\eta_{t-1}$; JP and US in $z_{t-1}$) appear to be more consistent as lagged one-period news are all positive and significant. This pattern reflects a market phenomenon, which shows that although the news has a negative effect on stock returns in the current period, the markets do rebound in the subsequent two periods. This pattern is also shown in the PALA markets, although the effect is not as uniform as it is in the G7 markets. Thus, the evidence in general supports the uncertainty premium hypothesis.

Third, the coefficients in the variance equation indicate that the GARCH(1,1) model in general is appropriate although some variations are found across different markets. Interestingly, testing results indicate that both $\Delta\eta_{t-1}$ and $\Delta z_{t-1}$ firmly contribute to explain variations in the variance, as evidenced by positive and significant coefficients for each market. The only exception is the Chinese market where no evidence exists to support the proposition that the variance in Chinese market volatility can be significantly predicated by EPU innovation or by its historical pattern. We further test the joint significance of all the lagged news variables in the system by setting the null as $\eta_{t-1} = \eta_{t-2} = z_{t-1} = z_{t-2} = 0$. The joint tests from the $\chi(4)$ indicate that the null is strongly rejected. Likewise, the joint test for $\Delta\eta_{t-1} = \Delta z_{t-1} = 0$ in the conditional variance by $\chi(2)$ also indicates the rejection of the null (except in the case in China); this evidence goes against the studies by Li (2017); Lopez de Carvalho (2017) and Chen et al. (2017), who fail to include the lagged EPU innovations in their models for predicting the conditional variance. The testing results thus conclude that the null hypothesis that stock returns are independent of lagged EPU innovations is rejected.

**Table 7.** Regression estimates of the G7 stock returns on domestic EPU and global EPU with GED-GARCH(1,1)-M procedure: January 1997–June 2016.

| Markets | $\alpha$ | $\eta_t$ | $\eta_{t-1}$ | $\eta_{t-2}$ | $z_t$ | $z_{t-1}$ | $z_{t-2}$ | $\rho_1$ | $\omega$ | $\varepsilon^2_{t-1}$ | $\sigma^2_{t-1}$ | $\Delta\eta_{t-1}$ | $\Delta z_{t-1}$ | $\chi(4)$ | $\chi(2)$ | $\bar{R}^2$ |
|---|---|---|---|---|---|---|---|---|---|---|---|---|---|---|---|---|
| CA | 1.887 | −0.038 | 0.012 | 0.017 | −0.088 | 0.068 | 0.014 | −0.006 | 17.410 | 0.116 | 0.343 | 0.241 | 0.514 | 1093.00 | 25.20 | 0.10 |
|  | 2.37 | −3.70 | 0.85 | 1.68 | −4.10 | 2.24 | 0.57 | −0.08 | 2.73 | 1.60 | 1.93 | 4.71 | 3.17 | [0.03] | [0.00] |  |
| FR | 0.377 | −0.025 | 0.020 | 0.003 | −0.086 | 0.056 | 0.011 | 0.014 | 0.269 | 0.017 | 0.963 | 0.080 | 0.320 | 16.33 | 34.49 | 0.13 |
|  | 0.65 | −4.48 | 2.23 | 0.33 | −4.07 | 2.22 | 0.47 | 0.20 | 1.13 | 0.99 | 40.10 | 2.07 | 5.32 | [0.00] | [0.00] |  |
| GM | −0.175 | −0.066 | 0.047 | 0.022 | −0.135 | 0.076 | 0.016 | −0.054 | 19.578 | 0.186 | 0.178 | 0.170 | 0.547 | 60.83 | 34.80 | 0.15 |
|  | −0.19 | −8.97 | 3.62 | 1.95 | −5.82 | 2.58 | 0.73 | −0.69 | 5.11 | 1.72 | 1.69 | 2.61 | 4.09 | [0.00] | [0.00] |  |
| IT | −0.024 | −0.054 | 0.034 | 0.021 | −0.081 | 0.080 | −0.011 | −0.021 | 1.641 | 0.159 | 0.788 | 0.153 | 0.293 | 42.26 | 16.71 | 0.14 |
|  | −0.02 | −5.33 | 2.81 | 1.90 | −7.16 | 4.05 | −0.63 | −0.27 | 1.69 | 2.72 | 11.23 | 2.66 | 2.55 | [0.00] | [0.00] |  |
| JP | 1.654 | −0.100 | 0.051 | 0.034 | −0.038 | 0.022 | 0.037 | −0.104 | 1.365 | 0.115 | 0.827 | 0.245 | 0.002 | 52.34 | 9.81 | 0.16 |
|  | 1.74 | −6.63 | 2.38 | 2.36 | −2.65 | 1.01 | 1.97 | −1.49 | 2.38 | 3.96 | 4.54 | 2.81 | 0.01 | [0.00] | [0.00] |  |
| UK | −1.211 | −0.015 | 0.022 | 0.001 | −0.106 | 0.067 | −0.003 | −0.091 | 4.946 | 0.220 | 0.472 | −0.040 | 0.166 | 55.39 | 6.96 | 0.12 |
|  | −3.23 | −3.98 | 4.70 | 0.16 | −6.78 | 2.99 | −0.15 | −1.33 | 2.64 | 2.11 | 2.50 | −1.47 | 2.47 | [0.00] | [0.03] |  |
| US | −0.348 | −0.036 | 0.001 | 0.039 | −0.029 | 0.024 | 0.032 | −0.097 | 1.537 | 0.046 | 0.868 | 0.317 | −0.097 | 51.76 | 59.55 | 0.07 |
|  | −0.40 | −4.86 | 4.90 | 5.64 | −1.30 | 0.96 | 1.24 | −1.15 | 2.32 | 1.20 | 16.00 | 7.32 | −1.20 | [0.00] | [0.00] |  |

Notes: The dependent variable is stock return. The values in the first row are the estimated coefficients and in the second row are the *t*-statistics. The $\eta_t$ is the domestic EPU at time *t*, and $z_t$ is the global EPU at time *t*. $\chi(4)$ is the chi-squared distribution for testing joint significance of lagged news in the mean equation. That is, $\eta_{t-1} = \eta_{t-2} = z_{t-1} = z_{t-2} = 0$. $\chi(2)$ is the chi-squared distribution for testing joint significance of $\Delta\eta_{t-1} = \Delta z_{t-1} = 0$ in the variance equation.

**Table 8.** Regression estimates of stock returns on domestic EPU and global EPU with GED-GARCH(1,1)-M procedure: Asian-Pacific and Latin American markets: January 1997–June 2016.

| Markets | $\alpha$ | $\eta_t$ | $\eta_{t-1}$ | $\eta_{t-2}$ | $z_t$ | $z_{t-1}$ | $z_{t-2}$ | $\rho_1$ | $\omega$ | $\varepsilon^2_{t-1}$ | $\sigma^2_{t-1}$ | $\Delta\eta_{t-1}$ | $\Delta z_{t-1}$ | $\chi(4)$ | $\chi(2)$ | $\bar{R}^2$ |
|---|---|---|---|---|---|---|---|---|---|---|---|---|---|---|---|---|
| AU | 2.052 | −0.024 | −0.021 | 0.033 | −0.028 | −0.018 | 0.039 | −0.168 | 0.345 | 0.110 | −0.195 | 0.145 | 0.032 | 18.41 | 7.85 | 0.13 |
|  | 4.24 | −3.05 | −1.82 | 3.76 | −1.96 | −1.010 | 2.71 | −2.28 | 0.12 | 1.008 | −0.55 | 2.80 | 0.42 | [0.00] | [0.02] |  |
| CN | 1.039 | −0.027 | 0.005 | 0.018 | −0.070 | 0.021 | −0.008 | 0.021 | 62.381 | 1.617 | 0.749 | −1.231 | −1.748 | 192.73 | 0.16 | 0.03 |
|  | 4.65 | −16.40 | 2.74 | 9.31 | −13.06 | 11.80 | −2.21 | 2.01 | 0.36 | 0.53 | 1.57 | −0.29 | −0.15 | [0.00] | [0.93] |  |
| HK | 1.563 | −0.042 | 0.009 | 0.025 | −0.052 | 0.016 | 0.019 | 0.102 | 1.849 | 0.078 | 0.861 | 0.001 | 0.345 | 12.77 | 9.27 | 0.10 |
|  | 1.62 | −5.04 | 1.23 | 3.06 | −2.47 | 0.52 | 0.79 | 1.05 | 2.32 | 1.44 | 13.98 | 0.02 | 2.43 | [0.01] | [0.01] |  |
| IN | 2.593 | −0.057 | 0.014 | 0.027 | 0.039 | −0.091 | 0.034 | 0.044 | 3.870 | 0.153 | 0.761 | 0.236 | 0.269 | 37.69 | 7.98 | 0.13 |
|  | 2.23 | −3.96 | 0.71 | 1.60 | 1.67 | −4.43 | 2.16 | 0.40 | 2.12 | 1.50 | 6.08 | 1.76 | 1.62 | [0.00] | [0.02] |  |
| KO | 2.566 | −0.071 | 0.054 | 0.002 | −0.056 | 0.049 | 0.052 | 0.040 | 14.088 | 0.350 | 0.548 | 0.514 | 0.057 | 19.22 | 25.69 | 0.01 |
|  | 1.59 | −5.60 | 3.06 | 0.10 | −1.83 | 1.00 | 1.61 | 20.50 | 3.97 | 4.03 | 9.23 | 5.01 | 0.16 | [0.00] | [0.00] |  |
| SG | 1.791 | −0.033 | −0.010 | 0.028 | 0.015 | −0.035 | 0.064 | 0.271 | 5.797 | 0.478 | 0.426 | 0.330 | −0.192 | 25.54 | 28.43 | 0.10 |
|  | 1.29 | −3.17 | −0.48 | 2.87 | 0.32 | −0.59 | 1.19 | 2.15 | 3.92 | 3.66 | 6.25 | 5.18 | −0.83 | [0.00] | [0.00] |  |
| BR | −1.477 | −0.020 | 0.017 | 0.015 | −0.089 | 0.066 | 0.024 | 0.022 | 19.723 | 0.497 | 0.348 | 0.211 | 0.545 | 14.40 | 10.49 | 0.02 |
|  | −1.86 | −1.82 | 1.26 | 1.89 | −2.87 | 1.71 | 0.89 | 0.21 | 2.49 | 4.82 | 2.47 | 3.17 | 2.00 | [0.01] | [0.01] |  |
| CL | 2.294 | −0.036 | −0.001 | 0.023 | −0.039 | 0.060 | −0.022 | 0.006 | 9.511 | 0.221 | 0.426 | 0.055 | 0.533 | 12.89 | 26.19 | 0.04 |
|  | 1.93 | −3.93 | −0.11 | 1.67 | −3.86 | 2.55 | −1.55 | 0.05 | 2.93 | 1.80 | 2.47 | 1.40 | 4.50 | [0.02] | [0.00] |  |

Notes: The dependent variable is stock return. The values in the first row are the estimated coefficients and in the second row are the *t*-statistics. The $\eta_t$ is the domestic EPU at time *t*, and $z_t$ is the global EPU at time *t*. $\chi(4)$ is the chi-squared distribution for testing joint significance of lagged news in the mean equation. That is, $\eta_{t-1} = \eta_{t-2} = z_{t-1} = z_{t-2} = 0$. $\chi(2)$ is the chi-squared distribution for testing joint significance of $\Delta\eta_{t-1} = \Delta z_{t-1} = 0$ in the variance equation.

## 6. Conclusions

This paper examines EMH and tests the news impact on stock returns by employing monthly data for 15 international equity markets. A simple way to test market efficiency is by examining the dependency of return series. By focusing on the univariate correlation analysis of stock returns, the statistics suggest that the null for the absence of correlations up to 12 months is rejected for 13 out of 15 markets; the exceptions are the U.S. and Canada. However, tests of the absence of autocorrelations of absolute values of stock returns are uniformly rejected for all markets under investigation.

We also test whether the news variables have significant effects on the stock returns. By using EPU indices as news variables, this study concludes that stock returns are negatively correlated with EPU in the current period, but are positively correlated in the following two periods, and the estimated coefficients are statistically significant in the majority of cases. This finding reflects a pattern of behavior among investors whose fears about the market, following bad news and the accompanying uncertainty, prompt them to sell off their stocks. This sell-off results in a fall in prices. However, rational traders may take advantage of declining prices and place orders, causing a bounce back in prices in the following two months. This phenomenon produces positive relations between stock returns and lagged news; this group of investors will receive uncertainty premiums, regardless of whether the news originates from a local market or the global market.

In placing the EPU innovations in the variance equation, the evidence consistently shows a predictive power in projecting stock volatility, not only using local news but also global lagged news. The only exception to this finding is the Chinese market, where we are unable to find a significant effect of EPU innovation in predicting variance. In sum, the evidence drawn from this study concretely shows that the news is significant in predicting future stock returns, which allows us to reject the EMH.

Since this study focuses on the time series dynamics to examine the EMH, the impact of accounting information on stock prices has been excluded from this study, but will be considered in future study by factoring in the quality of financial reporting along the line of Ohlson (1995); Glezakos et al. (2012) and Jianu et al. (2014).

**Funding:** This research was partly funded by the Marshal M. Austin Endowed Chair established in March 1996, Le Bow College of Business, Drexel University.

**Conflicts of Interest:** The author declares no conflicts of interest.

## Appendix A

**Table A1.** Summary of notations of variables.

| Variable | Description | Source |
|---|---|---|
| $p_t = ln(P_t)$ | $P_t$ is the market stock index for each country. | Datastream |
| $R_{m,t}$ | Market stock returns, which is obtained by taking the natural log-difference of stock price index times 100. | Datastream |
| $\phi_{m,t-1}$ | Market information set up to time $t-1$. | |
| $\sigma_t^2$ | Variance of stock returns generated from the GARCH(1,1)-M process | |
| $\rho_s$ | Autocorrelation coefficient with s period lag. | |
| $\eta_{t-i}$ | Economic policy uncertainty index at time $t-i$ from Baker et al. (2016). This variable was transformed by taking the natural logarithm. | Baker et al. (2016) * |
| $\Delta\eta_{t-1}$ | EPU innovation measured by natural log-difference of the EPU index. | |
| $z_{t-i}$ | Global economic policy uncertainty index at time $t-i$ from Davis (2016). This variable was transformed by taking the natural logarithm. | Davis (2016) * |

**Table A1.** *Cont.*

| Variable | Description | Source |
| --- | --- | --- |
| $\Delta z_{t-1}$ | Global EPU innovation measured by the natural log-difference of GEPU index. | |
| $\varepsilon_t$ | Random error term. | |
| $\Omega_{t-1}$ | Information set conditional on time $t-1$ in the empirical test. | |
| CED $(\cdot)$ | Generalized error distribution. | |
| G7 | Group 7 industrial markets | |
| APLA | Asian-Pacific and Latin American (APLA) markets | |

\* http://www.policyuncertainty.com Website information has been updated to the current. Source: 'Measuring Economic Policy Uncertainty' by Scott Baker, Nicholas Bloom and Steven J. Davis at www.PolicyUncertainty.com.

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
