# Peer review of "Market Efficiency and News Dynamics: Evidence from International Equity Markets"

_economies, doi:10.3390/economies7010007_

Round 1
Reviewer 1 Report
Referee report for the paper “Market Efficiency and the Timing of News Arrival: Evidence from
International Equity Markets”
Summary of the paper
This paper studies market efficiency and the impact of news on stock returns in 15 international
equity markets, including the stock market on G7 and Asian-Pacific and Latin American (APLA)
countries. The authors study the autocorrelation structure of stock returns at the monthly level,
and they also study the impact of economic policy uncertainty (EPU) on the stock return and its
volatility at the monthly level. The authors find some significant autocorrelations and also show
that EPU may help predict stock returns. Based on these results, they conclude that markets are
not efficient.
Main Comment
While I am sympathetic with the idea and methods used in the paper, unfortunately I could not
find any significant contribution with respect to the existing literature. In particular, Lopes de
Carvalho (2017) has already done a very similar analysis for G7 countries, and she does it for
both stocks and bonds: in a similar setting and with a much deeper analysis.1
Lopes de Carvalho (2017) finds that EPU has great influence on stock returns and conditional
volatility, for most of the countries in her sample. Moreover, she also finds that previous stock
returns are significant in explaining the current stock returns estimation for many countries.
Interestingly, when analyzing bonds she finds that the influence of EPU on current bond returns
is much lower.
Other comments
Section 2 on the Theory of Market Efficiency should be an appendix, in my view.
Page 7 describes Figures 3 and 4, however, there are no Figures 3 and 4 in the paper.
Correlation tables may be more useful than figures to establish comovements.
1 The article is available in this link: https://repositorio.iscte-iul.pt/bitstream/10071/15713/1/teresa_lopes_carvalho_diss_mestrado.pdf
2
The title for Table 6 should contain APLA instead of G7.
The content on the last column on Tables 2 and 3 remains a mystery, as it is not described in the
text or captions of the paper.
Conclusion
The model presented does not include a significant contribution with respect to Lopes de
Carvalho (2017). The presentation of the results could be improved. Many of the results
presented are not new.
References
Lopez de Carvalho, T., 2017, “Economic policy uncertainty and return on financial assets – The
G7 Case.” Master in Finance dissertation at ISCTE Business School. Available at
https://repositorio.iscte-iul.pt/bitstream/10071/15713/1/teresa_lopes_carvalho_diss_mestrado.pdf
Author Response
See the attached file for my reply. Thanks!

Reviewer 2 Report
Dear Author(s)
Thank you for an interesting study.
Please, read the comments below that in my opinion will significantly improve your article;
Abstract:
In my opinion abstract should be of more informative nature. There are a lot of very detailed information – technical words but they make the abstract unclear for the potential readers. First of all the aim of the paper should be emphasized, then applied methods. As a next part the author should briefly present his/her results and explains contribution and value added of the paper and recommendation for practice or further study.
Introduction:
The introduction is too detailed, especially at the beginning is „stuffed” with numbers. It makes the paper unclear from the beginning. I understand the intentions of the author, although it should be more calm and keep the typical elements of the introduction: justification of the choice of the topic but without too detailed information (details will be in the empirical part), the purpose of the article, a description of the research method used and a brief description of each section of the paper.
In my opinion the aim of the research should be more emphasized, starting with “the aim of the paper is….”. The introduction should be completed with brief description of applied methods.
References – Citations:
When giving citation there is a need to indicate very precisely who and where said/wrote it, thus pages should be added to in-text references e.g. „Fama (1976) wrote … „ please add p. where?
Empirical part:
For the sake of readability of this part, I propose that the variables used should be presented in tabular form with explanations of where, what function will be used.
Findings and discussion:
It is obvious that author of the paper is a specialist in the methodological part. It is unquestionable advantage of the paper but there is a lacks theoretical foundations. In my opinion, the article should be completed with a thorough literature review. In the basis of theoretical assumption and literature review research hypotheses can be introduced and verify in the study – this is definitely missing.
There is also a lack of discussion. It is recommended to confront the findings with the results of other authors.
Conclusions:
Conclusions should be enrich with limitations of the study as well as recommendations for practice and future research.
I suggest to divide the article into the following parts and proceed according this structure: Introduction, Theoretical foundations - literature review and hypotheses development, Materials and Methods, Results and discussion, Conclusions.
Reviewer 3 Report
the paper presents major flaws and inconsistencies in the research design, research questions and the execution and presentation of the main findings.
Author Response
Revisions were made based on other reviewers.
Reviewer 4 Report
Recommendations:
· All the abbreviations used in the formula should be explained. The abbreviations have to be explained the first time when they are used
· All the amounts presented in the tables have to be explained.
· Because the time data are used in the models, the mandatory tests to validate the results have to be included in the article.
· Are all of the capital markets used in this study efficient? Is there a difference between the efficient and emerging capital markets? Give explanation in the article about this aspect.
· Using only EPU as variable to predict the stock price is not enough. You miss the most important variables: financial information; solve the problems with omitted variables
· Page 1: Too many amounts in the introduction. Express clearly the idea.
· Page 2: “First, as news is released, regardless of whether its focus is financial or political, its impact on the markets via market participants’ reassessments of their portfolio positions is immediate” – what are the sources here? Also the market reacts at changes in regulations (example: “The value relevance of financial reporting in Romania”, Economic Computation and Economic Cybernetics Studies and Research, vol. 48, no .4, pp. 167-182). I recommend to include this statement in your article.
· Page 3: “This also informs investors that if there are deviations by using ??−1 in predicting future stock price, ??, the errors must be associated with random news that hits the market between time t-1 and t.” This is not true. The Ohlson model used in many articles, including in the article mentioned above, proved that information from financial statements are the main sources in predicting the stock price.
· Page 4: APLA not ABLA
· Page 4: In the text, the author mention that the data are for the period January 1997 – June 2016, but in the Table 1: the summary statistics are for the period 1990.09 – 2016.06. What is the reality? Why do you choose 310 (265, 263) observations? How did you arrive at this number of observations?
· Page 5: What do the axes in the Figure 1 and 2 represent?
· Page 6: How do you expect that we understand the explanations based on the figures 3 and 4 which are missing from the paper?
· Page 11: In the tables 5, 6, 7, 8 are the t-statistics significant? Haw many observations are used to test the models?
I expect in the revised version to see an article written in a professional way.
Author Response
See attached file for reply.

Round 2
Reviewer 1 Report
Referee report for the paper “Market Efficiency and the Timing of News Arrival: Evidence from
International Equity Markets”: Second round
The paper has improved notably due to a significant effort in trying to explain the main contribution
of the paper, in relation with the existing literature. I applaud the effort.
Comments
1. As I mentioned in my previous report, the paper by Lopes de Carvalho (2017) is very closely
related. While I appreciate that you now cite her paper, I believe that, given how close it is to
the one under revision, it deserves a separate paragraph in the introduction, in which you
should highlight what your marginal contribution is with respect to her paper. The content of
the paragraph should be similar to your response (point 6) to my previous report.
2. As I also mentioned in my previous report, Lopes de Carvalho (2017) analyzes a similar
question to yours for both stocks and bonds. Could you please incorporate bonds in your
empirical analysis? This would significantly improve your paper, providing results on
different and major asset classes.
3. In the paper under review, you mention that your data goes from January 1997 through June
2016. Why stop there and not go all the way through December 2018 (or as recently as
possible)? Markets have moved very much since 2016: Brexit, the Trump election, the tax
reform, trade wars, geopolitical risks, etc. Indeed, even in your introduction you provide
motivating examples about market movements in 2018. Why not include those in your study
by adding the most recent data?
Other comments
In the paper, the authors cite Lopes de Carvalho (2017) as “de Carvalho (2017).” I believe the
surname is Lopes de Carvalho and it should be cited as “Lopes de Carvalho (2017)”.
References
Lopez de Carvalho, T., 2017, “Economic policy uncertainty and return on financial assets – The G7
Case.” Master in Finance dissertation at ISCTE Business School. Available at
https://repositorio.iscte-iul.pt/bitstream/10071/15713/1/teresa_lopes_carvalho_diss_mestrado.pdf
Author Response
Thanks your comments.
· The last name of Lopez de Carvalho, T. (2017) was corrected.
· Explanations of data presented in the tables 2,3 and 4 were added. Particularly, the first row is the estimated coefficients and the second row reports the t-statistics.
· I added: * indicates statistically significant at the 5% level.
· The footnote of Table 4 made clear for testing autocorrelations of the absolute with 12 lags by, the null is uniformly rejected at the 1% level for all markets as shown in P-values.
Reviewer 2 Report
The current state of the article satisfies me. I accept in the current version.
Author Response
Thanks for your comments. I made additional changes as follows:
The last name of Lopez de Carvalho, T. (2017) was corrected.
Explanations of data presented in the tables 2,3 and 4 were added. Particularly, the first row is the estimated coefficients and the second row reports the t-statistics.
I added: * indicates statistically significant at the 5% level.
The footnote of Table 4 made clear for testing autocorrelations of the absolute with 12 lags by, the null is uniformly rejected at the 1% level for all markets as shown in P-values.
Reviewer 4 Report
The article confirms the results of the last studies that questioning the efficiency of capital market.
In order to be easily understandable for the readers, I recommend to explain clearly the data presented in the tables 2, 3 and 4 (explain in the notes what the amounts in the first and second line mean - e.g estimated coefficients and the t-statistics). Also mention what is the level of significance for the sign “*” in the tables 2 and 3.
Author Response
Thanks for your comments. I should not miss the information. The corrections are as follows.
The last name of Lopez de Carvalho, T. (2017) was corrected.
Explanations of data presented in the tables 2,3 and 4 were added. Particularly, the first row is the estimated coefficients and the second row reports the t-statistics.
I added: * indicates statistically significant at the 5% level.
The footnote of Table 4 made clear for testing autocorrelations of the absolute with 12 lags by, the null is uniformly rejected at the 1% level for all markets as shown in P-values.
Round 3
Reviewer 1 Report
I provided three main comments and one minor comment. The authors decided to ignore my main comments and pursue only my minor comment, without even providing an explanation of why they did not incorporate any of the main comments.